# State-dependent cell-type-specific membrane potential dynamics and unitary synaptic inputs in awake mice

Aurélie Pala[1,2]*, Carl CH Petersen[1]*

[1]Laboratory of Sensory Processing, Brain Mind Institute, Faculty of Life Sciences, École Polytechnique Fédérale de Lausanne (EPFL), Lausanne, Switzerland; [2]Wallace H. Coulter Department of Biomedical Engineering, Georgia Institute of Technology and Emory University, Atlanta, United States

**Abstract** The cellular and synaptic mechanisms driving cell-type-specific function during various cortical network activities and behaviors are poorly understood. Here, we targeted whole-cell recordings to two classes of inhibitory GABAergic neurons in layer 2/3 of the barrel cortex of awake head-restrained mice and correlated spontaneous membrane potential dynamics with cortical state and whisking behavior. Using optogenetic stimulation of single layer 2/3 excitatory neurons we measured unitary excitatory postsynaptic potentials (uEPSPs) across states. During active states, characterized by whisking and reduced low-frequency activity in the local field potential, parvalbumin-expressing neurons depolarized and, albeit in a small number of recordings, received uEPSPs with increased amplitude. In contrast, somatostatin-expressing neurons hyperpolarized and reduced firing rates during active states without consistent change in uEPSP amplitude. These results further our understanding of neocortical inhibitory neuron function in awake mice and are consistent with the hypothesis that distinct genetically-defined cell classes have different state-dependent patterns of activity.

DOI: https://doi.org/10.7554/eLife.35869.001

*For correspondence:
aurelie.pala@gmail.com (AP);
carl.petersen@epfl.ch (CCHP)

**Competing interests:** The authors declare that no competing interests exist.

## Introduction

In cortical excitatory neurons, reduced low-frequency neocortical EEG or local field potential (LFP) activity, such as that observed during active behaviors, consistently correlates with a decrease in membrane potential ($V_m$) variance through a reduction in the amplitude of low-frequency $V_m$ fluctuations, accompanied, on average, by $V_m$ depolarization (*Steriade et al., 2001*; *Timofeev et al., 2001*; *Crochet and Petersen, 2006*; *Poulet and Petersen, 2008*; *Yamashita et al., 2013*; *Bennett et al., 2013*; *Polack et al., 2013*; *Schneider et al., 2014*; *Reimer et al., 2014*; *Zhao et al., 2016*). Much less is known regarding $V_m$ and its relationship to action potential (AP) firing in inhibitory GABAergic neurons during various states of awake neocortical activity and during active behaviors. In layer 2/3 (L2/3) mouse primary somatosensory whisker barrel cortex (wS1), fast-spiking inhibitory neurons were found to decrease AP firing during whisking with little change in mean $V_m$ but decreased $V_m$ variance (*Gentet et al., 2010*). In comparison, parvalbumin-expressing (PV) neurons in L2/3 mouse visual cortex exhibited $V_m$ depolarization accompanied by an increase in AP firing during locomotion (*Polack et al., 2013*). Disparities also exist amongst somatostatin-expressing (Sst) neurons, which are inhibited during whisking in L2/3 of wS1 (*Gentet et al., 2012*; *Lee et al., 2013*; *Muñoz et al., 2017*), and either excited or inhibited by locomotion in L2/3 visual cortex (*Polack et al., 2013*; *Reimer et al., 2014*).

Mechanistically, changes in synaptic efficacy could contribute to driving state-dependent $V_m$ dynamics, but in vivo measurements of synaptic transmission have largely been carried out under

anesthesia (*Matsumura et al., 1996*; *Crochet et al., 2005*; *Bruno and Sakmann, 2006*; *Jouhanneau et al., 2015*; *Pala and Petersen, 2015*; *Safari et al., 2017*; *Jouhanneau et al., 2018*). Further experiments are therefore needed to investigate the cellular and synaptic mechanisms contributing to cell-type-specific and state-dependent $V_m$ dynamics during wakefulness and active behaviors. Here, we carried out whole-cell recordings to measure $V_m$ fluctuations in PV and Sst neurons in L2/3 of wS1 in awake head-restrained mice, and, in the subset of synaptically connected recordings, we probed unitary synaptic inputs through optogenetic stimulation of single nearby excitatory neurons.

## Results

### $V_m$ dynamics in PV and Sst neurons across cortical and behavioral states

We made two-photon targeted whole-cell recordings of PV (*Figure 1A*) (*Hippenmeyer et al., 2005*) and Sst (*Figure 1B*) (*Taniguchi et al., 2011*) neurons expressing tdTomato (*Madisen et al., 2010*) in L2/3 of the C2 barrel column of wS1 in awake, head-restrained mice. Simultaneously, we recorded nearby LFP and conducted high speed filming of whisker movements to define cortical states and whisking-related behavioral states. We identified periods of high and low LFP power in the 1–5 Hz frequency band, as the amplitude of low frequency activity is known to correlate with various levels of arousal and to be modulated by movement (*Steriade, 2000*; *Gervasoni et al., 2004*; *Crochet and Petersen, 2006*; *McGinley et al., 2015*). Epochs with and without whisking were determined based on the velocity of the C2 whisker (*Figure 1A,B*) (see Materials and methods).

Cortical states with high 1–5 Hz LFP power largely occurred during non-whisking periods, whereas whisking periods were dominated by low 1–5 Hz LFP power (*Figure 2—figure supplement 2*). In further analyses, we therefore distinguished the two predominant non-overlapping states: Quiet periods defined as epochs with high 1–5 Hz LFP power without whisker movement, and Active periods defined as epochs with low 1–5 Hz LFP power accompanied by whisker movement. During Active states, PV neurons depolarized (*Figure 2A*), reduced $V_m$ standard deviation (*Figure 2B*), and reduced AP firing rate (*Figure 2C*). During Active states, PV neurons also reduced the amplitude of slow-frequency $V_m$ fluctuations (*Figure 2D,E*), and decreased $V_m$ *vs* LFP cross-correlation (*Figure 2F*). In contrast, $V_m$ of Sst neurons hyperpolarized during Active states (*Figure 2A*) without a change in $V_m$ standard deviation (*Figure 2B*), giving rise to a reduced AP firing rate (*Figure 2C*). Sst neurons had low amplitude slow-frequency $V_m$ fluctuations during both Quiet and Active states (*Figure 2D,E*), and $V_m$ of Sst neurons showed little correlation with LFP, irrespective of state (*Figure 2F*). The $V_m$ differences between PV and Sst neurons were not due to overall differences in cortical states or whisking-related behavior across different genotypes of mice (*Figure 2—figure supplement 1*). Separate analyses of whisking-related and cortical state-related $V_m$ modulation suggested that PV neurons may be relatively more strongly modulated by cortical state, whereas Sst neurons may be relatively more strongly modulated by whisking (*Figure 2—figure supplements 2* and *3*).

Altogether, these results show a cell-type-specific modulation of $V_m$ across cortical and behavioral states in L2/3 wS1 of awake head-restrained mice. AP firing rates reduced during Active states in both PV and Sst neurons, but through distinct changes in $V_m$ dynamics.

### Excitatory unitary synaptic inputs in PV and Sst neurons across cortical and behavioral states

Enhanced efficacy of local excitatory synaptic input onto PV neurons could contribute to the overall depolarization of PV neurons in Active states, and equally decreased efficacy of excitatory synaptic input onto Sst neurons could contribute to the overall hyperpolarisation of Sst neurons in Active states. We tested these specific hypotheses by measuring unitary synaptic inputs across states. Through two-photon targeted single-cell electroporation (*Kitamura et al., 2008*; *Pala and Petersen, 2015*), we expressed a fast channelrhodopsin, *Chronos* (*Klapoetke et al., 2014*), in a single L2/3 excitatory neuron per mouse (*Figure 3—figure supplement 1*). Brief pulses of light (1 ms, 1 Hz) delivered using a 470 nm LED elicited highly reliable, time-locked single APs at short latency in *Chronos*-expressing neurons across both Quiet and Active states (*Figure 3A–C*, *Figure 3—figure supplement 1*), evoking unitary excitatory postsynaptic potentials (uEPSPs) in PV (*Figure 3D–F*) and

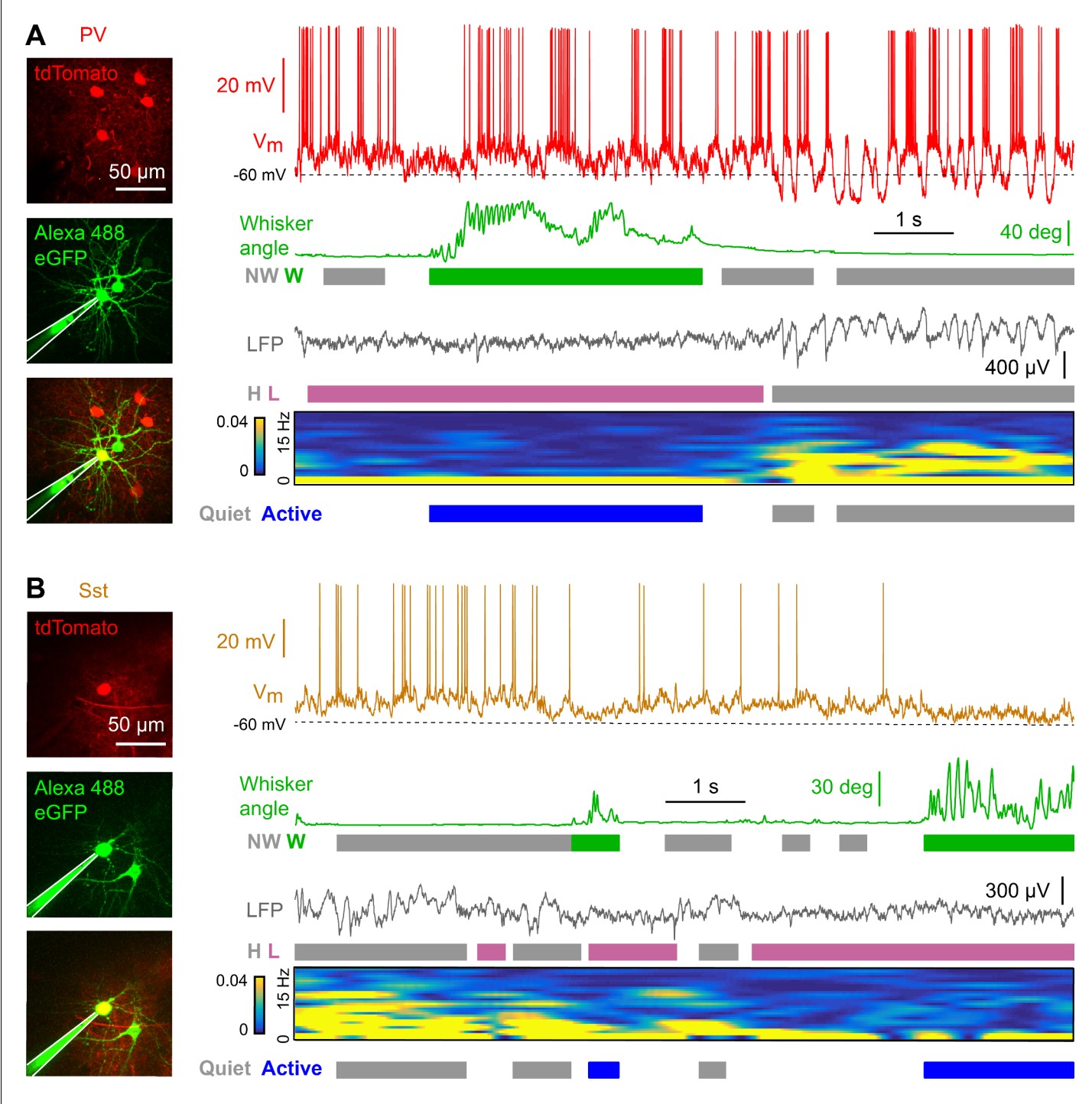

**Figure 1.** Membrane potential recordings of PV- and Sst-expressing GABAergic neurons in layer 2/3 of the awake mouse barrel cortex with simultaneous measurement of whisker position and local field potential. (**A**) Example recording of a PV-expressing neuron. From top to bottom: Membrane potential ($V_m$), whisker angle, local field potential (LFP), normalized LFP FFT power. Green/grey boxes represent Whisking/Not-Whisking states, pink/grey color boxes represent Low/High 1–5 Hz LFP power states, and blue/grey boxes represent Active/Quiet states. (**B**) Same as in panel A, but for a Sst-expressing neuron.

DOI: https://doi.org/10.7554/eLife.35869.002

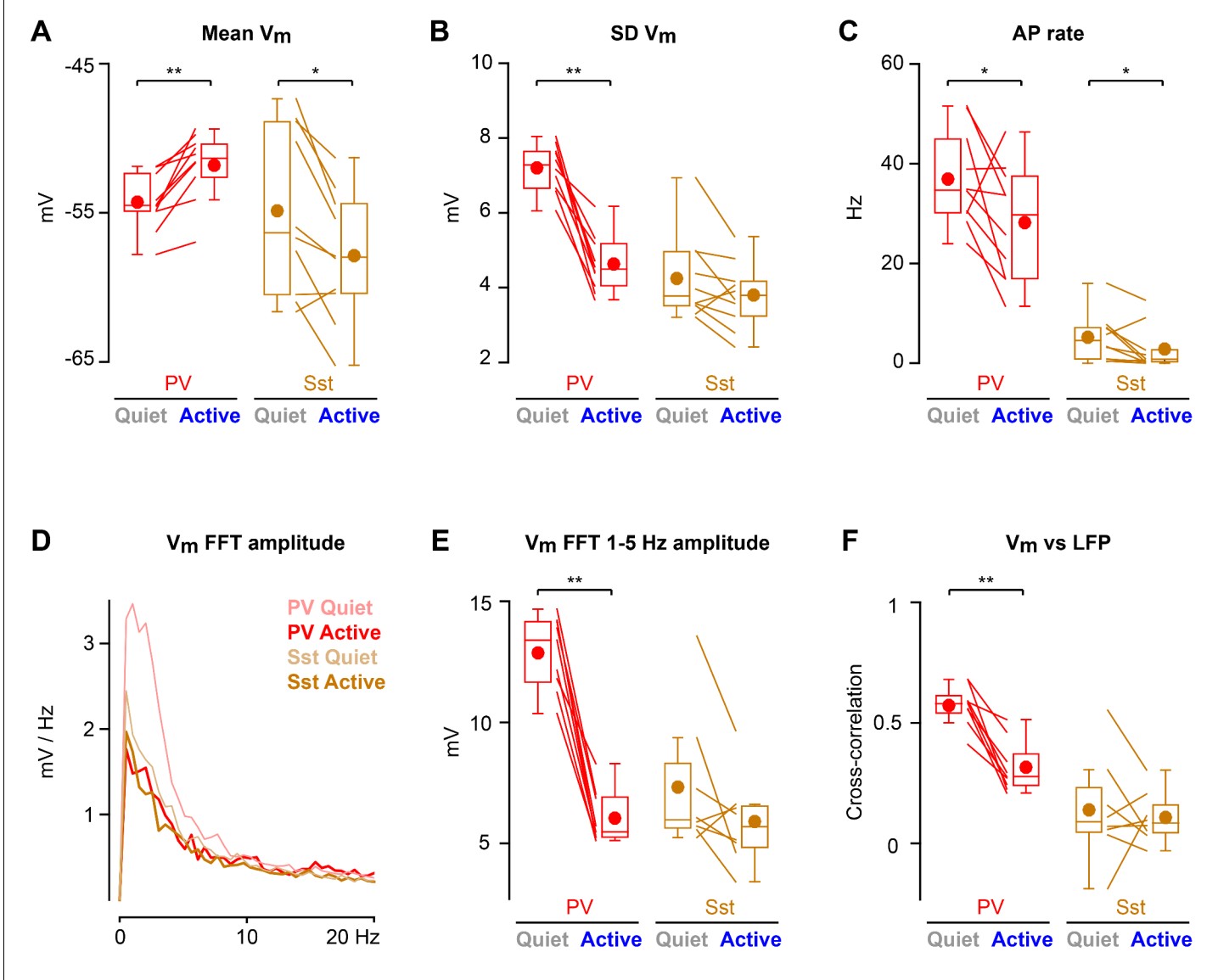

**Figure 2.** State-dependent modulation of membrane potential dynamics in PV and Sst neurons. (**A**) Mean membrane potential ($V_m$). (**B**) Standard deviation (SD) of $V_m$. (**C**) Spontaneous action potential (AP) rate. (**D**) $V_m$ FFT amplitude spectrum. (**E**) $V_m$ FFT amplitude in the 1–5 Hz frequency band. (**F**) Peak cross-correlation between $V_m$ and LFP. Two-tailed Wilcoxon signed-rank test assessed statistical significance, with ** indicating $p<0.01$ and * indicating $p<0.05$.

DOI: https://doi.org/10.7554/eLife.35869.003

The following source data and figure supplements are available for figure 2:

**Source data 1.** Data values and statistics underlying *Figure 2*.
DOI: https://doi.org/10.7554/eLife.35869.007

**Figure supplement 1.** LFP dynamics and whisking behavior are similar in PV-Cre x LSL-tdTomato and Sst-Cre x LSL-tdTomato.
DOI: https://doi.org/10.7554/eLife.35869.004

**Figure supplement 2.** Differential modulation of membrane potential dynamics in PV and Sst neurons by cortical state and whisking behavior.
DOI: https://doi.org/10.7554/eLife.35869.005

**Figure supplement 3.** Membrane potential dynamics of PV neurons apear to be more strongly modulated by cortical state, whereas Sst neurons appear to be more strongly modulated by whisking behavior.
DOI: https://doi.org/10.7554/eLife.35869.006

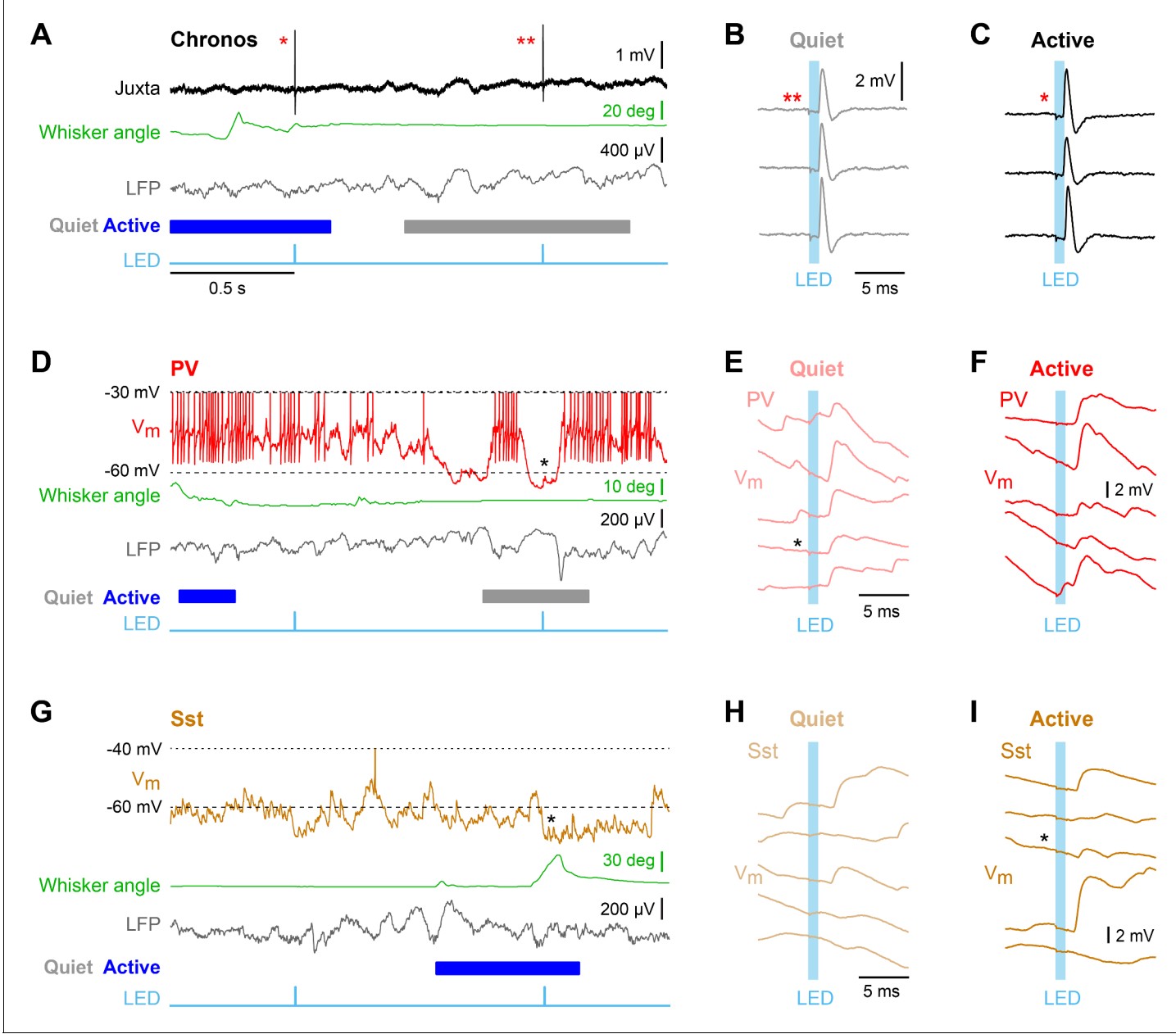

**Figure 3.** Unitary excitatory postsynaptic potentials in PV and Sst neurons in layer 2/3 of the awake mouse barrel cortex measured together with whisker position and local field potential. (**A**) Example juxtacellular recording of a presynaptic *Chronos*-expressing neuron. From top to bottom: Extracellular signal (Juxta), whisker angle, LFP, and light stimulus (LED). Blue/grey color boxes represent Active and Quiet states. (**B and C**) Time-locked individual APs evoked by a 1 ms LED stimulus during Quiet and Active states. (**D**) Example whole-cell recording from a PV neuron together with whisker angle, LFP and LED stimulus. (**E and F**) Individual uEPSP responses to 1 ms optogenetic stimuli during Quiet and Active states. (**G–I**) Same as panels D-F, but for a Sst neuron.

DOI: https://doi.org/10.7554/eLife.35869.008

The following figure supplement is available for figure 3:

**Figure supplement 1.** Precisely-evoked single action potentials in excitatory neurons expressing *Chronos*.

DOI: https://doi.org/10.7554/eLife.35869.009

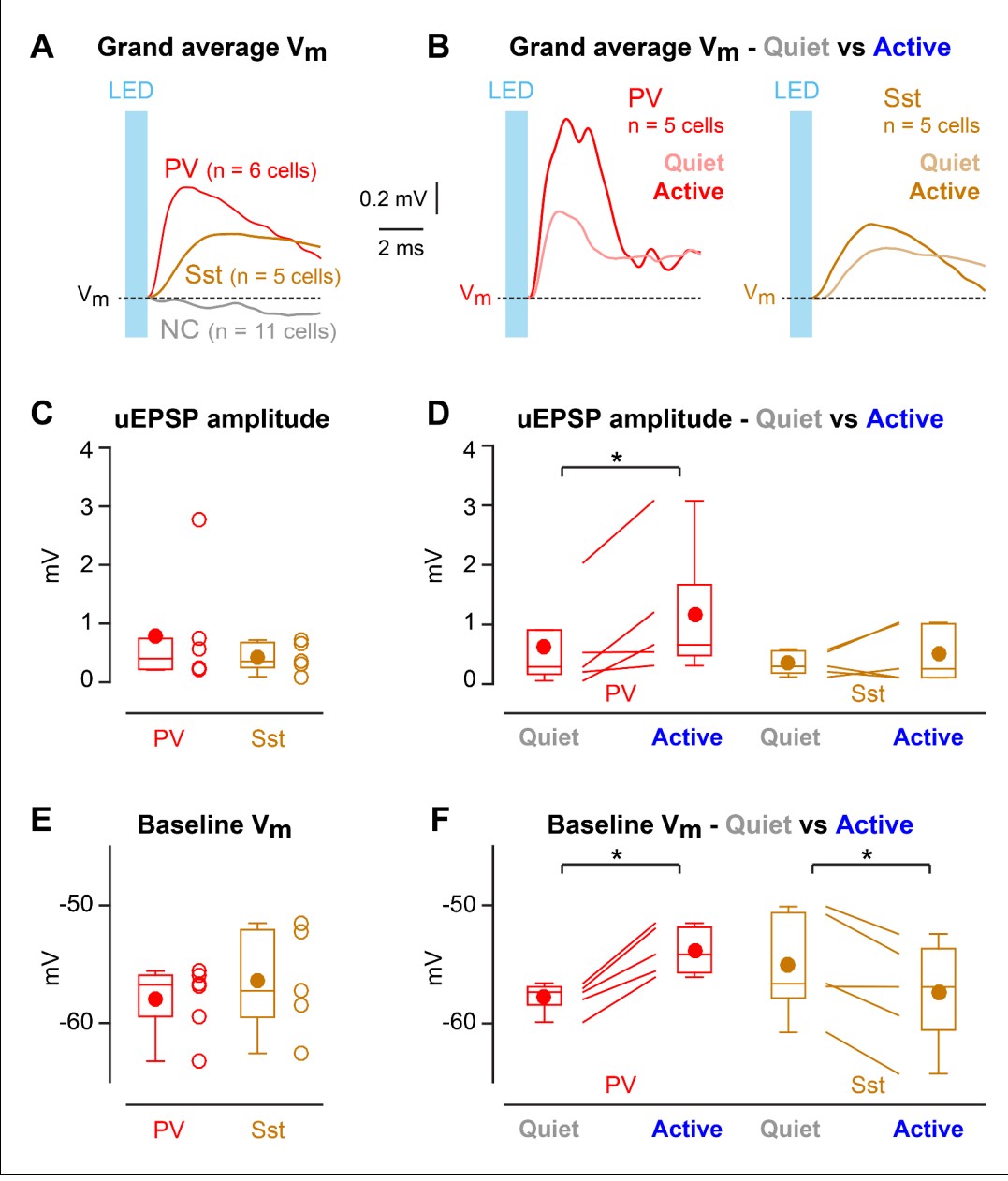

**Figure 4.** State-dependent modulation of excitatory synaptic input in PV neurons. (**A**) Mean optogenetically-evoked $V_m$ responses for PV, Sst, and all non-connected (NC) neurons. (**B**) Mean uEPSPs evoked in PV and Sst neurons during Quiet and Active states. (**C**) uEPSP amplitudes across PV and Sst neurons. (**D**) uEPSP amplitudes in PV and Sst neurons during Active and Quiet states. (**E**) Baseline $V_m$ at uEPSP onset in PV and Sst neurons. (**F**) Baseline $V_m$ at uEPSP onset in PV and Sst neurons during Active and Quiet states. Two-tailed Wilcoxon rank-sum test assessed statistical significance for panels C and E, and none was found. One-tailed Wilcoxon signed-rank test assessed statistical significance for panels D and F, with * indicating $p < 0.05$.

DOI: https://doi.org/10.7554/eLife.35869.010

The following source data and figure supplement are available for figure 4:

**Source data 1.** Data values and statistics underlying *Figure 4*.
DOI: https://doi.org/10.7554/eLife.35869.012

**Figure supplement 1.** Kinetics of excitatory synaptic input in PV and Sst neurons.
DOI: https://doi.org/10.7554/eLife.35869.011

Sst neurons (*Figure 3G–I*). The uEPSPs were faster in PV neurons than in Sst neurons, but with overall similar amplitudes (*Figure 4A,C Figure 4—figure supplement 1*). In the small number of synaptically-connected postsynaptic neurons, we found an increase in uEPSP amplitude in PV neurons (n = 5 cells, p=0.031, one-tailed Wilcoxon signed-rank test) during Active states compared to Quiet states, but no consistent change in uEPSP amplitude in Sst neurons (n = 5 cells, p=0.22, one-tailed Wilcoxon signed-rank test) (*Figure 4B,D*). In the Active state, the baseline $V_m$ (from which uEPSPs were evoked) was more depolarized for PV neurons, but more hyperpolarized for Sst neurons (*Figure 4F*), in agreement with the overall state-dependent $V_m$ changes (*Figure 2A*). Altogether, these results suggest an enhancement of local excitatory synaptic transmission onto PV neurons during Active states.

## Discussion

Our measurements in L2/3 of wS1 reveal prominent cell-type-specific differences in $V_m$ dynamics across cortical states and whisking behavior. Mechanistically, modulation of local unitary synaptic input strength might contribute to these state-dependent $V_m$ dynamics.

### Sst-expressing GABAergic neurons

Sst neurons hyperpolarized (*Figure 2A*) and reduced firing rates (*Figure 2C*) during Active states, in agreement with previous measurements of whisking-related modulation in L2/3 of wS1 (*Gentet et al., 2012*; *Lee et al., 2013*; *Muñoz et al., 2017*). The hyperpolarization of Sst neurons is thought to be driven by increased firing of VIP-expressing GABAergic neurons, which strongly inhibit Sst neurons (*Lee et al., 2013*; *Pfeffer et al., 2013*; *Pi et al., 2013*). The reduction in AP firing rate of Sst neurons during Active states may disinhibit distal dendrites of nearby excitatory neurons (*Gentet et al., 2012*), perhaps promoting non-linear dendritic excitation important for sensorimotor integration and perceptual decision-making (*Xu et al., 2012*; *Takahashi et al., 2016*).

Interestingly, the $V_m$ of Sst neurons only showed low-amplitude slow fluctuations (*Figure 2D,E*) and little correlation with LFP (*Figure 2F*), neither of which were affected by Quiet vs Active states. Sst neurons in L2/3 mouse barrel cortex are therefore functionally relatively uncoupled from the surrounding neuronal network, within which all other cell-types show highly correlated low frequency $V_m$ fluctuations (*Poulet and Petersen, 2008*; *Gentet et al., 2010*; *Gentet et al., 2012*). In addition to prominent inhibitory input from VIP neurons, Sst neurons receive excitatory synaptic input, which shows strong frequency-dependent facilitation (*Reyes et al., 1998*; *Kapfer et al., 2007*; *Silberberg and Markram, 2007*; *Pala and Petersen, 2015*). Short-term synaptic plasticity together with cell-type- and layer-specific circuits likely contribute to the relatively decoupled $V_m$ dynamics of Sst neurons compared to other types of nearby neurons.

Nicotinic enhancement of uEPSPs has recently been reported on Sst neurons in wS1 of anesthetised mice (*Urban-Ciecko et al., 2018*), and acetylcholine is known to be released during whisking (*Eggermann et al., 2014*). However, in our study with limited sample size, optogenetically-evoked uEPSPs in Sst neurons had similar amplitude during Quiet and Active epochs, suggesting comparable local excitatory drive across states. Future studies with a larger sample size, and with additional classification of subtypes of Sst neurons (*Muñoz et al., 2017*), may well reveal state-dependent synaptic transmission onto Sst neurons in awake mice. Furthermore, it is possible that diverse synaptic inputs (for example from different sources such as from thalamus, or different cell-types in different layers of various cortical regions) onto Sst neurons might be differentially modulated by diverse cortical and behavioral states, and in future experiments it will be particularly important to measure synaptic transmission during execution and learning of goal-directed behaviors.

### PV-expressing GABAergic neurons

$V_m$ recordings from PV cells revealed that these fast-spiking GABAergic neurons are strongly modulated across Active and Quiet states. During Quiet states PV neurons were on average hyperpolarized (*Figure 2A*) and exhibited large $V_m$ variance (*Figure 2B*) in the slow frequency range (*Figure 2D,E*) highly correlated with the LFP (*Figure 2F*). Conversely, during Active states, the $V_m$ of PV neurons depolarized together with a reduction in $V_m$ variance, $V_m$ slow fluctuations and $V_m$ correlation with LFP. Despite depolarization, the AP firing rate of PV neurons reduced significantly during the Active state (*Figure 2C*), presumably because the decreased $V_m$ variance prevented an increased

frequency of $V_m$ excursions above AP threshold. Consistent with these observations, decreased firing of fast-spiking neurons was also previously noted in L2/3 of wS1 during whisking (*Gentet et al., 2010*), and during licking events accompanied by whisking in a whisker detection task (*Sachidhanandam et al., 2016*). These results appear to suggest that a reduction in the firing rate of PV neurons in L2/3 of wS1 typically accompanies whisking. The reduced firing rates of PV neurons during Active states will presumably disinhibit the surrounding neuronal network, perhaps promoting synaptic computations amongst the excitatory pyramidal neurons. Studies with larger sample size considering subtypes of PV neurons and comparing across various behavioral conditions will be important to further our understanding.

Optogenetic stimulation of a single nearby excitatory pyramidal neuron appeared to evoke uEPSPs with increased amplitude during Active states across the small sample of synaptically-connected postsynaptic PV neurons in our data set (*Figure 4B,D*). An increased amplitude of incoming L2/3 excitatory synaptic input could contribute to driving the depolarized average $V_m$ in PV neurons during the Active cortical state (*Figures 2A* and *4F*). Depolarization reduces the electrical driving force for glutamatergic conductances and therefore cannot explain increased amplitude of uEPSPs. In future experiments, it will be of interest to investigate whether state-dependent changes in the input resistance, or other aspects of dendritic integration, as well as the concentration of diverse neuromodulators might play a role in regulating synaptic efficacy through various presynaptic and postsynaptic mechanisms. Importantly, PV neurons receive synaptic inputs from many sources, which could be differentially regulated giving rise to complex state dependent $V_m$ dynamics. Understanding the mechanisms regulating synaptic transmission during behavior remains an important challenge, necessary for a causal understanding of cortical circuit function.

## Materials and methods

### Animal preparation, surgery, and habituation to head-restraint

Six 5–10 week old female and male PV-IRES-Cre (*Hippenmeyer et al., 2005*) mice and five 5–10 week old female and male Sst-IRES-Cre (*Taniguchi et al., 2011*) mice crossed with CAG-Lox-STOP-Lox-tdTomato (LSL-tdTomato) mice (*Madisen et al., 2010*) were used in accordance with protocols approved by the Swiss Federal Veterinary Office (authorisation VD1628). Mice were maintained under 1–2% isoflurane anesthesia while being implanted with a custom-made head-holder and a recording chamber. The location of the left C2 barrel column was functionally identified through intrinsic signal optical imaging under 0.5–1% isoflurane anesthesia (*Lefort et al., 2009*). Mice were habituated to head- and paw-restraint under different light conditions for 3–5 days before proceeding to electroporation and electrophysiological recordings.

### Single-cell electroporation

Mice were kept under 1% isoflurane anesthesia while a small craniotomy (diameter 1–1.5 mm) was made leaving the dura intact. Electroporation of a single non-tdTomato neuron per PV-Cre x LSL-tdTomato or Sst-Cre x LSL-tdTomato mouse was carried out as previously described (*Kitamura et al., 2008*; *Pala and Petersen, 2015*). In brief, a glass pipette with a resistance of 10–17 MΩ was filled with the same solution used for whole-cell recordings (see below) to which Alexa 488 dye (50–100 µM) (Invitrogen), and plasmid DNA pAAV-Syn-Chronos-eGFP (100 ng/µl) (kindly provided by Thomas Oertner) (*Klapoetke et al., 2014*) were added. A two-photon microscope (Prairie Technologies) was used to visualize the pipette and the tdTomato-negative cell somas as dark shadows over a brighter background. The pipette was inserted in the brain through the intact dura and brought into close contact with the cell body of the target neuron and 50 pulses of negative voltage step (0.5 ms, –12 V) were delivered at 50 Hz using a pulse generator (Axoporator 800A, Molecular Devices). The craniotomy was then covered with silicone elastomer (Kwik-Cast, WPI) and the mice were returned to their home cage for 24 hr before proceeding to electrophysiological recordings.

### Electrophysiology

24 hr after electroporation, mice were re-anesthetized with 1–2% isoflurane and the dura was removed. A drop of agarose gel (1.2% in Ringer solution) (Sigma) was placed on top of the

craniotomy, which was then partially sealed with a coverslip (#1 thickness, Menzel-Gläser) held in place with cyanoacrylate glue (Loctite, Henkel). The recording chamber was filled with Ringer solution and capped with silicone elastomer (Kwik-Cast, WPI). Mice were returned to their home cage and left to recover from anesthesia for a minimum of 2 hr. Mice were head-restrained under the two-photon microscope, the silicone elastomer cap removed and the agarose gel cleared from the non-sealed part of the craniotomy. The location of the single *Chronos*-expressing neuron was identified by cortical blood vasculature pattern and its excitatory nature was confirmed by its overall morphology and the presence of numerous dendritic spines. Local field potential (LFP) was continuously recorded with a 2–4 MΩ glass pipette filled with Ringer solution containing 10–25 µM Alexa 594 dye and lowered to a depth of 150–250 µm below the pia and within 250 µm from the *Chronos*-expressing neuron. Two-photon targeted juxtacellular recording of the *Chronos*-expressing neuron was performed with 5–7 MΩ glass pipettes filled with the same solution as used for LFP recordings. Two-photon targeted whole-cell patch-clamp recordings were performed as previously described (*Margrie et al., 2003*; *Gentet et al., 2010*; *Yamashita et al., 2013*; *Pala and Petersen, 2015*). 5–8 MΩ glass pipettes were filled with a solution containing (in mM): 135 potassium gluconate, 4 KCl, 10 HEPES, 10 sodium phosphocreatine, 4 MgATP, 0.3 $Na_3GTP$ (adjusted to pH 7.3 with KOH), to which 25–75 µM Alexa 488 dye was added. Patch-clamp recordings were obtained in current-clamp mode without current injection and $V_m$ was not corrected for liquid junction potentials. All recorded signals were amplified by a Multiclamp 700B amplifier (Axon Instruments), Bessel filtered at 10 kHz and digitized at 20 kHz by an ITC-18 (Instrutech Corporation) under the control of custom written routines in IgorPro (Wavemetrics).

### Optogenetic stimulation

A collimated 470 nm superbright LED (Luxeon, Philips) was placed at the back of the 40x/0.8NA two-photon objective (Olympus) to generate wide field stimulation. Optogenetic stimuli consisted of single square pulses of light of 1 ms duration and 58 $mW/mm^2$ intensity, delivered at a frequency of 1 Hz. A constant 470 nm background illumination made of an array of small LEDs (Everlight Electronics) was located in front of the mouse and kept on for the duration of the whole recording session.

### Whisker filming

On recording day, all whiskers except for the left and right C2 whiskers were trimmed. Whisker movements were filmed at 200 Hz (CamRecord CL600 × 2, Optronis) with a resolution of 13 pixels/mm using the 470 nm LED array (see above) as an illumination source. During one postsynaptic $V_m$ recording, we failed to acquire high-speed filming data due to a disk error.

### Data analysis

Epochs of whisking (W) and not-whisking (NW) were identified according to the speed of whisker movement using custom routines written in ImageJ (NIH) and Matlab (MathWorks). Portions of the recordings not assigned to either W or NW categories were not considered in the analysis. Epochs of high and low LFP power in the 1–5 Hz frequency band were identified for each recorded *Chronos*-expressing neuron and each postsynaptic neuron. Briefly, for each recorded trial of spontaneous and optogenetically-evoked activity, the LFP was down-sampled to 2000 Hz, low-pass filtered at 200 Hz (forward and reverse direction) and the power in the 1–5 Hz band was computed using a sliding FFT (window size: 2 s, overlap: 10 ms). A distribution of the 1–5 Hz power values was generated for each recorded neuron and portions of the recording were assigned to 'Low 1–5 Hz LFP power (L)' if their corresponding FFT power values were smaller than the 40th percentile of the distribution. Similarly, portions of the recording were assigned to 'High 1–5 Hz LFP power (H)' if their corresponding FFT power values were larger than the 60th percentile of the distribution. The 'Active' state was then defined as periods of recording displaying Low 1–5 Hz LFP power together with whisking, while the 'Quiet' state was defined as periods with High 1–5 Hz LFP power without whisker movement.

Electrophysiological properties of PV and Sst neurons were quantified as follows. Mean $V_m$ and standard deviation (SD) of $V_m$ were computed for spontaneous periods of recording excluding APs. The $V_m$ *vs* LFP cross-correlation was computed for segments of spontaneous activity of 1 s duration. The $V_m$ was offset by its average value and normalized by its standard deviation and the LFP was low-pass filtered at 200 Hz. To compute the $V_m$ FFT spectrum and FFT amplitude in the 1–5 Hz

frequency band, the $V_m$ was median-filtered to remove APs. Segments of spontaneous activity of 1 s duration were used to compute the $V_m$ FFT.

To quantify the light-evoked AP responses of the presynaptic *Chronos*-expressing neurons, an AP was considered as optogenetically evoked if its peak occurred within 10 ms of the onset of the 1 ms light stimulus. AP latency was defined as the time elapsed between light stimulus onset and AP peak time. AP jitter was defined as the standard deviation of the AP latency.

To quantify the properties of the light-evoked uEPSPs, an optogenetic stimulus-triggered smoothed $V_m$ average was computed, excluding stimuli with postsynaptic APs occurring in a 50 ms window starting 20 ms before stimulus onset (*Pala and Petersen, 2015*). We analyzed 54 ± 32 (mean ± SD, median 50) stimuli during the Quiet state (n = 10 cells) and 32 ± 18 (34) stimuli during the Active state (n = 10 cells). uEPSP amplitude was calculated as the difference between the mean $V_m$ averaged over a 0.25 ms window centered at the peak of the uEPSP and the mean baseline $V_m$ averaged over a 0.25 ms window taken immediately prior to the onset of the uEPSP. The uEPSP onset latency was defined as the time at which the smoothed first derivative of the $V_m$ exceeded a threshold of 100 mV/s for PV neurons and 60 mV/s for Sst neurons. The uEPSP peak was defined as the first time at which the smoothed first derivative of the $V_m$ became negative post uEPSP onset. The uEPSP rise time corresponded to the time elapsed from 20% to 80% of the amplitude on the rising phase of the averaged uEPSP. Classification of an optogenetic stimulus in a given category (Quiet/Active) required that 20 ms prior to stimulus onset and 30 ms post stimulus onset continuously be assigned to that category.

Population data are represented as mean ± SD in bar plots. In box plots, the median and interquartile range are shown with whiskers extending from the smallest data point comprised within 1.5x the interquartile range of the 1$^{st}$ quartile to the largest data point comprised within 1.5x the interquartile range of the third quartile. The mean value is superimposed on the box plots with a filled circle. Two-tailed Wilcoxon rank-sum and signed-rank tests were used to compare two groups of unpaired and paired data respectively, except in *Figure 4D,F* where we applied one-tailed Wilcoxon signed-rank tests. We justify use of one-tailed statistics in *Figure 4D*, because we test the specific hypothesis of whether uEPSP amplitude increased during Active states in PV neurons, thus contributing to their depolarization. For Sst neurons, we test the specific hypothesis of whether uEPSP amplitude decreased during Active states, thus contributing to their hyperpolarization. In *Figure 4F*, we justify use of one-tailed statistics because we test the specific hypotheses that PV neurons depolarize during Active states (as already found in *Figure 2A*) and that Sst neurons hyperpolarize during Active states (as already found in *Figure 2A*). Data analysis and statistical analysis were carried out in Matlab (Mathworks).

### Data availability

The complete data set and Matlab analysis code are freely available at the CERN database Zenodo (https://zenodo.org/communities/petersen-lab-data) with DOI: https://doi.org/10.5281/zenodo.1304771.

## Acknowledgements

We thank Thomas Oertner for providing the plasmid DNA encoding *Chronos*; Taro Kiritani for help with whisker filming and for sharing ImageJ plugins for movie analysis; and Sylvain Crochet and Bilal Haider for comments on an earlier version of the manuscript. This work was supported by grants from the Swiss National Science Foundation and the European Research Council.

## Additional information

### Funding

| Funder | Grant reference number | Author |
| --- | --- | --- |
| Swiss National Science Foundation | 310030B_166595 | Carl CH Petersen |
| European Research Council | ERC-ADG-293660 | Carl CH Petersen |

The funders had no role in study design, data collection and interpretation, or the decision to submit the work for publication.

## Author contributions
Aurélie Pala, Conceptualization, Software, Formal analysis, Investigation, Methodology, Writing—original draft; Carl CH Petersen, Conceptualization, Resources, Supervision, Funding acquisition, Writing—original draft

## Author ORCIDs
Aurélie Pala (iD) https://orcid.org/0000-0002-9910-8470
Carl CH Petersen (iD) http://orcid.org/0000-0003-3344-4495

## Ethics
Animal experimentation: All experiments were carried out in accordance with protocols approved by the Swiss Federal Veterinary Office (authorisation VD1628). Six 5–10-week-old female and male PV-IRES-Cre (Hippenmeyer et al., 2005) mice and five 5–10-week-old female and male Sst-IRES-Cre (Taniguchi et al. 2011) mice crossed with CAG-Lox-STOP-Lox-tdTomato (LSL-tdTomato) mice (Madisen et al., 2010) were used in accordance with protocols approved by the Swiss Federal Veterinary Office.

## Decision letter and Author response
Decision letter https://doi.org/10.7554/eLife.35869.019
Author response https://doi.org/10.7554/eLife.35869.020

# Additional files

## Supplementary files
• Transparent reporting form
DOI: https://doi.org/10.7554/eLife.35869.013

## Data availability
The complete data set and Matlab analysis code are freely available at the CERN database Zenodo (https://zenodo.org/communities/petersen-lab-data) with DOI: https://doi.org/10.5281/zenodo.1304771.

The following dataset was generated:

| Author(s) | Year | Dataset title | Dataset URL | Database, license, and accessibility information |
|---|---|---|---|---|
| Pala A, Petersen CCH | 2018 | Data set for "State-dependent cell-type-specific membrane potential dynamics and unitary synaptic inputs in awake mice" | https://doi.org/10.5281/zenodo.1304771 | Publicly available at Zenodo (https://zenodo.org/communities/petersen-lab-data), Creative Commons, Open Access |

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
