## [Decision Letter]

Thank you for submitting your article "Cell-type-specific modulation of membrane potential dynamics and unitary synaptic inputs by behavior and cortical state" for consideration by *eLife*. Your article has been reviewed by three peer reviewers, including Yang Dan as the Reviewing Editor and Reviewer #1, and the evaluation has been overseen by Eve Marder as the Senior Editor.

The reviewers have discussed the reviews with one another and the Reviewing Editor has drafted this decision to help you prepare a revised submission.

In this study, the authors make targeted patch-clamp recordings to examine the excitatory synaptic inputs to parvalbumin-expressing (PV) and somatostatin-expressing (SOM) interneurons in the barrel cortex of awake mice during different cortical states and whisking behaviors. They optogenetically activated single excitatory neurons and measured the evoked uEPSPs. They showed that during desynchronized cortical state, PV-expressing interneurons are depolarized and receive larger uEPSPs, whereas during whisking Sst neurons are hyperpolarized but showed no change in uEPSPs.

Previous studies from this group of investigators showed distinct effects of brain and behavioral states on the firing rates of PV and Sst neurons. By measuring the excitatory inputs into each subtype, the current study provides new insights into the synaptic mechanism underlying the firing rate modulation. In addition, they found differential effects of cortical state vs. whisking behavior on modulating the membrane potential dynamics of PV and Sst neurons, which will no doubt inspire future studies on the underlying mechanism. Methodologically, the approach builds on their 2015 paper (Pala and Petersen, 2015) which addressed similar questions, but here they carry out the experiments in awake mice while tracking LFP and whisker movements. These are highly challenging experiments and the recordings are of high quality. The dataset is valuable and the results are potentially interesting for our understanding of synaptic integration in defined interneuron populations in the awake cortex, though the authors could to more to explain the significance of their findings to a wider audience.

Essential revisions:

- The n's are generally quite low which might lead to erroneous conclusions (see e.g. Figure 4D, PV Q vs W is not significant – even though 4/5 cells go up. The same can be seen in more graphs in Figure 4). Given that Figure 4 provides the only really novel results in this manuscript – attempting to link what is already known in the literature about how interneurons react to changes in state and whisking to the excitatory input that they receive from local pyramidal cells – and recognizing that these experiments are extremely difficult – if a few additional n's could be added to the dataset in Figure 4 it would greatly strengthen the study. If this is not practical, the authors should perhaps tone down their claims to reflect the limitations of the data.

- A related concern is on the conclusion that "PV neurons are modulated more strongly by cortical state, while Sst neurons are modulated more strongly by whisking behavior". Looking at Figure 2, although there are differences in the magnitude and significance levels of the modulation by cortical state and whisking, the trends tend to be in the same direction. This raises some question as to whether the difference is sensitive to how H and L LFP powers are defined. Are the observed differences robust to whether the threshold is set at 40th percentile? Since in Figure 1C, lower 1-5 Hz LFP power occurs in both Q and W behavioral states, can the authors compare the PV and Sst activity between Q and W within the low LFP state? In addition. the authors claim that PV neurons do not change firing when cortical state changes – however, Figure 1 shows how the slow LFP fluctuations changes the PV firing to a burst-like firing pattern. Additional analyses to demonstrate the robustness of the result will be helpful.

- A major concern of the analysis that has been done is that the "whisking" state and the "LOW 1-5Hz" state are obviously strongly correlated (as shown in Figure 1). However, it seems that they did not correct for this when analysing their data (making sure that the amount of whisking is matched when comparing cortical states and vice versa). Therefore, it is unclear, if e.g. the reduction in 1-5Hz power in PV neurons during whisking is actually due to the whisking or because the whisking periods coincide with "Low 1-5Hz" periods which correlate with a reduced 1-5Hz power in PV neurons (Figure 2D). The same problem applies to all plots in Figure 2 and Figure 4C-J.

- The single-cell electroporation technique is elegant but also tricky. Based on the loose-patch access, there could be leaking and labeling of neurons nearby. Only one example image is shown in Figure 1. Some quantification and more images for the recordings would be important to demonstrate the unitary stimulation. It would also be important to show how stable the recording of the same neuron is, whether there is any change of the access in different brain or behavioral states.

- The unitary excitatory input examined in this study is the intralaminar (l2/3) local input. The major excitatory inputs for a L2/3 PV neuron could be from L4 or thalamic neurons, and it could be different for Sst neurons. It would be very helpful for the authors to discuss how these different inputs could shape the membrane potential dynamics of l2/3 PV and Sst inhibitory neurons respectively.

- The authors include "behaviour" in the title, leading the reader to expect that the animal was actively engaged in some task, but the authors were simply referring to "whisking or not". They should consider removing 'behaviour' from the title.

- The Discussion section sticks very close to the data and the authors give almost no consideration to how their findings relate to cortical function and current models of cortical processing of sensory input. This should be addressed.

---

## [Author Response]

In this study, the authors make targeted patch-clamp recordings to examine the excitatory synaptic inputs to parvalbumin-expressing (PV) and somatostatin-expressing (SOM) interneurons in the barrel cortex of awake mice during different cortical states and whisking behaviors. They optogenetically activated single excitatory neurons and measured the evoked uEPSPs. They showed that during desynchronized cortical state, PV-expressing interneurons are depolarized and receive larger uEPSPs, whereas during whisking Sst neurons are hyperpolarized but showed no change in uEPSPs.Previous studies from this group of investigators showed distinct effects of brain and behavioral states on the firing rates of PV and Sst neurons. By measuring the excitatory inputs into each subtype, the current study provides new insights into the synaptic mechanism underlying the firing rate modulation. In addition, they found differential effects of cortical state vs. whisking behavior on modulating the membrane potential dynamics of PV and Sst neurons, which will no doubt inspire future studies on the underlying mechanism. Methodologically, the approach builds on their 2015 paper (Pala and Petersen, 2015) which addressed similar questions, but here they carry out the experiments in awake mice while tracking LFP and whisker movements. These are highly challenging experiments and the recordings are of high quality. The dataset is valuable and the results are potentially interesting for our understanding of synaptic integration in defined interneuron populations in the awake cortex, though the authors could to more to explain the significance of their findings to a wider audience.

We thank the reviewers for their insightful comments. We have rewritten the Discussion section to help explain implications of our results. However, the manuscript is submitted as a Short Report, preventing lengthy Discussion section.

Essential revisions:- The n's are generally quite low which might lead to erroneous conclusions (see e.g. Figure 4D, PV Q vs W is not significant – even though 4/5 cells go up. The same can be seen in more graphs in Figure 4). Given that Figure 4 provides the only really novel results in this manuscript – attempting to link what is already known in the literature about how interneurons react to changes in state and whisking to the excitatory input that they receive from local pyramidal cells – and recognizing that these experiments are extremely difficult – if a few additional n's could be added to the dataset in Figure 4 it would greatly strengthen the study. If this is not practical, the authors should perhaps tone down their claims to reflect the limitations of the data.

Unfortunately, carrying out new experiments is not feasible, since the first author, who carried out all the experimental work, has moved from the EPFL to Georgia Tech. These experiments are very difficult to perform, and, unfortunately, at the moment there is no one in our laboratory trained in these techniques to carry out further such experiments. We agree that n numbers are low, and that we should tone down our conclusions. We have therefore made important changes to the manuscript, including new analyses and explicitly pointing out important limitations in our study in terms of sample size:

Abstract: “During active states, characterized by whisking and reduced low-frequency activity in the local field potential, parvalbumin-expressing neurons depolarized and, albeit in a small number of recordings, received uEPSPs with increased amplitude.”

Subsection “Sst-expressing GABAergic neurons”: “However, in our study with limited sample size, optogenetically-evoked uEPSPs in Sst neurons had similar amplitude during Quiet and Active epochs, suggesting comparable local excitatory drive across states. Future studies with a larger sample size, and with additional classification of subtypes of Sst neurons (Muñoz et al., 2017), may well reveal state-dependent synaptic transmission onto Sst neurons in awake mice. Furthermore, it is possible that diverse synaptic inputs (for example from different sources such as from thalamus, or different cell-types in different layers of various cortical regions) onto Sst neurons might be differentially modulated by diverse cortical and behavioral states, and in future experiments it will be particularly important to measure synaptic transmission during execution and learning of goal-directed behaviors.”

In subsection “PV-expressing GABAergic neurons”: “The reduced firing rates of PV neurons during Active states will presumably disinhibit the surrounding neuronal network, perhaps promoting synaptic computations amongst the excitatory pyramidal neurons. Studies with larger sample size considering subtypes of PV neurons and comparing across behavioral conditions will be important to further our understanding.

Optogenetic stimulation of a single nearby excitatory pyramidal neuron evoked uEPSPs with increased amplitude during Active states across the small sample of synaptically-connected postsynaptic PV neurons in our data set (Figure 4B,D). An increased amplitude of incoming L2/3 excitatory synaptic input could contribute to driving the depolarized average V_m_ in PV neurons during the Active cortical state (Figures 2A and 4F). Depolarization reduces the electrical driving force for glutamatergic conductances and therefore cannot explain increased amplitude of uEPSPs. In future experiments, it will be of interest to investigate whether state-dependent changes in the input resistance, or other aspects of dendritic integration, as well as the concentration of diverse neuromodulators might play a role in regulating synaptic efficacy through various presynaptic and postsynaptic mechanisms. Importantly, PV neurons receive synaptic inputs from many sources, which could be differentially regulated giving rise to complex state dependent V_m_ dynamics. Understanding the mechanisms regulating synaptic transmission during behavior remains an important challenge, necessary for a causal understanding of cortical circuit function.”

- A related concern is on the conclusion that "PV neurons are modulated more strongly by cortical state, while Sst neurons are modulated more strongly by whisking behavior". Looking at Figure 2, although there are differences in the magnitude and significance levels of the modulation by cortical state and whisking, the trends tend to be in the same direction. This raises some question as to whether the difference is sensitive to how H and L LFP powers are defined. Are the observed differences robust to whether the threshold is set at 40th percentile? Since in Figure 1C, lower 1-5 Hz LFP power occurs in both Q and W behavioral states, can the authors compare the PV and Sst activity between Q and W within the low LFP state? In addition. the authors claim that PV neurons do not change firing when cortical state changes – however, Figure 1 shows how the slow LFP fluctuations changes the PV firing to a burst-like firing pattern. Additional analyses to demonstrate the robustness of the result will be helpful.

We agree with the reviewers, and we have now made important changes in the analysis. The main figures of the manuscript now focus on a comparison of the two predominant cortical and behavioral states: Quiet periods defined as epochs with high 1-5 Hz LFP power without whisker movement, and Active periods defined as epochs with low 1-5 Hz LFP power accompanied by whisker movement. The main story of the manuscript now compares cell-type-specific differences in V_m_ dynamics, V_m_ vs LFP correlations, and uEPSPs across these Quiet and Active periods.

We also carried out the further analyses suggested by the reviewers regarding the relative prominence of the modulation of PV and Sst neurons by cortical state versus whisking. In Figure 2—figure supplement 3, we compare the V_m_ dynamics and LFP vs V_m_ correlation between periods of Whisking and Non-Whisking occurring solely during the Low 1-5 Hz LFP power. We find that none of the metrics used to quantify PV neuron activity are changed by whisking behavior, whereas the mean V_m_ of Sst neurons is significantly more hyperpolarized during Whisking compared to Non-Whisking epochs.

Similarly, we compare periods of Low and High 1-5 Hz LFP power during Non-Whisking epochs and find that PV neuron V_m_ and V_m_ vs LFP correlation are significantly modulated by changes in cortical state in the absence of whisker movements, but not Sst neurons. These two new analyses plus our initial analysis, which we now show in Figure 2—figure supplement 2, therefore provide further support for differential cortical state vs whisking-related modulation of PV and Sst neurons. On P. 4 we write: “Separate analyses of whisking-related and cortical state-related V_m_ modulation suggested that PV neurons may be relatively more strongly modulated by cortical state, whereas Sst neurons may be relatively more strongly modulated by whisking (Figure 2—figure supplement 2 and Figure 2—figure supplement 3).”

We investigated whether changes in analysis thresholds affected our results, by changing the LFP 1-5 Hz power threshold from 40^th^ percentile to 33^rd^ percentile. We recomputed all analyses shown in Figure 2 and found the results to be robust.

The pattern of PV firing indeed changes comparing Quiet and Active states. During Quiet states the V_m_ fluctuates on a slow time-scale and action potentials are only fired during the depolarized epochs. Following the reviewer’s comment, we computed the inter-spike interval (ISI), to examine burst firing of action potentials within 50 ms of each other, which seems to be prominent in both states, see Author response image 1:

**Author response image 1 respfig1:** 

- A major concern of the analysis that has been done is that the "whisking" state and the "LOW 1-5Hz" state are obviously strongly correlated (as shown in Figure 1). However, it seems that they did not correct for this when analysing their data (making sure that the amount of whisking is matched when comparing cortical states and vice versa). Therefore, it is unclear, if e.g. the reduction in 1-5Hz power in PV neurons during whisking is actually due to the whisking or because the whisking periods coincide with "Low 1-5Hz" periods which correlate with a reduced 1-5Hz power in PV neurons (Figure 2D). The same problem applies to all plots in Figure 2 and Figure 4C-J.

We agree with the reviewers. As described above, we have changed our analyses, such that the main figures of the manuscript now focus on a comparison of the two predominant cortical and behavioral states: Quiet periods defined as epochs with high 1-5 Hz LFP power without whisker movement, and Active periods defined as epochs with low 1-5 Hz LFP power accompanied by whisker movement. The main story of the manuscript now compares cell-type-specific differences in V_m_ dynamics, LFP vs V_m_ correlations, and uEPSPs across these Quiet and Active periods.

In parallel, we carried out further analyses to better tease apart the contribution of cortical state and whisking behavior to the modulation of PV and Sst neurons. Figure 2—figure supplement 3 shows the results of two additional analyses, which focus on periods of Whisking and Non-Whisking during Low 1-5 Hz LFP Power only, and on periods of High and Low 1-5 Hz LFP Power during epochs lacking whisker movements. The results of such analyses seem to indicate a stronger modulation of PV neuron by cortical state compared to whisking, whereas Sst neurons are less affected by changes in cortical state while being more sensitive to whisker movements. The results of these new analyses plus our initial analysis (now reported in Figure 2—figure supplement 2) are mentioned in subsection “Vm dynamics in PV and Sst neurons across cortical and behavioral states” of the manuscript: “Separate analyses of whisking-related and cortical state-related modulation suggested that PV neurons may be relatively more strongly modulated by cortical state, whereas Sst neurons may be relatively more strongly modulated by whisking (Figure 2—figure supplement 2 and Figure 2—figure supplement 3).”

- The single-cell electroporation technique is elegant but also tricky. Based on the loose-patch access, there could be leaking and labeling of neurons nearby. Only one example image is shown in Figure 1. Some quantification and more images for the recordings would be important to demonstrate the unitary stimulation. It would also be important to show how stable the recording of the same neuron is, whether there is any change of the access in different brain or behavioral states.

To perform single-cell electroporation, we use glass pipettes with a tip diameter of ~1 µm and resistance of 10-17 MΩ, which have been shown to preferentially label single neurons (see Haas et al., 2001 for the first use of micropipettes to deliver DNA to individual neurons in vivo). Furthermore, in every experiment, we carefully visualized the electroporated area using the 2-photon microscope and verified that only a single neuron was transfected before proceeding to the recordings. There were no experiments in which there was any doubt about the number of presynaptic neurons. We have extensive experience in this technique (Yamashita et al., 2013; Pala and Petersen, 2015), and have carried out careful anatomical investigations (Yamashita et al., 2018). In Figure 3—figure supplement 1, we now show four additional example fields of view, with single-electroporated neurons.

We did not perform current injections throughout the recording to compare across Active and Quiet states, so we do not have a direct measure of access resistance. However, action potential half-width is a relatively sensitive measure of changes in access resistance. We therefore computed the action potential half-width across states and found no difference, see Author response image 2.

**Author response image 2. respfig2:** 

- The unitary excitatory input examined in this study is the intralaminar (l2/3) local input. The major excitatory inputs for a L2/3 PV neuron could be from L4 or thalamic neurons, and it could be different for Sst neurons. It would be very helpful for the authors to discuss how these different inputs could shape the membrane potential dynamics of l2/3 PV and Sst inhibitory neurons respectively.

The reviewer is correct to point out the importance of many diverse synaptic connections, all of which likely contribute to driving V_m_ dynamics. We now explicitly point this out:

Subsection “Sst-expressing GABAergic neurons”: “Future studies with a larger sample size, and with additional classification of subtypes of Sst neurons (Muñoz et al., 2017), may well reveal state-dependent synaptic transmission onto Sst neurons in awake mice. Furthermore, it is possible that diverse synaptic inputs (for example from different sources such as from thalamus, or different cell-types in different layers of various cortical regions) onto Sst neurons might be differentially modulated by diverse cortical and behavioral states, and in future experiments it will be particularly important to measure synaptic transmission during execution and learning of goal-directed behaviors.”

Subsection “PV-expressing GABAergic neurons”: “Studies with larger sample size considering subtypes of PV neurons and comparing across behavioral conditions will be important to further our understanding.”

Subsection “PV-expressing GABAergic neurons”: “Importantly, PV neurons receive synaptic inputs from many sources, which could be differentially regulated giving rise to complex state dependent V_m_ dynamics. Understanding the mechanisms regulating synaptic transmission during behavior remains an important challenge, necessary for a causal understanding of cortical circuit function.”

- The authors include "behaviour" in the title, leading the reader to expect that the animal was actively engaged in some task, but the authors were simply referring to "whisking or not". They should consider removing 'behaviour' from the title.

We have removed ‘behavior’ from the title.

- The Discussion section sticks very close to the data and the authors give almost no consideration to how their findings relate to cortical function and current models of cortical processing of sensory input. This should be addressed.

We have rewritten the Discussion section and tried to add ideas about the potential functional significance of our findings. We hope the reviewers find the manuscript to be improved.